# Research Progress on Transorgan Regulation of the Cardiovascular and Motor System through Cardiogenic Exosomes

**DOI:** 10.3390/ijms23105765

**Published:** 2022-05-21

**Authors:** Haoyang Gao, Lingli Zhang, Zhikun Wang, Kai Yan, Linlin Zhao, Weihua Xiao

**Affiliations:** Shanghai Frontiers Science Research Base of Exercise and Metabolic Health, Shanghai University of Sport, Shanghai 200438, China; haoyanggao1999@163.com (H.G.); lingliwdc@163.com (L.Z.); wangzhikun2131@163.com (Z.W.); yankai_1658925503@163.com (K.Y.)

**Keywords:** heart, exosomes, transorgan regulation, micro-RNA

## Abstract

The heart is the core organ of the circulatory system. Through the blood circulation system, it has close contact with all tissues and cells in the body. An exosome is an extracellular vesicle enclosed by a phospholipid bilayer. A variety of heart tissue cells can secrete and release exosomes, which transfer RNAs, lipids, proteins, and other biomolecules to adjacent or remote cells, mediate intercellular communication, and regulate the physiological and pathological activities of target cells. Cardiogenic exosomes play an important role in regulating almost all pathological and physiological processes of the heart. In addition, they can also reach distant tissues and organs through the peripheral circulation, exerting profound influence on their functional status. In this paper, the composition and function of cardiogenic exosomes, the factors affecting cardiogenic exosomes and their roles in cardiovascular physiology and pathophysiology are discussed, and the close relationship between cardiovascular system and motor system is innovatively explored from the perspective of exosomes. This study provides a reference for the development and application of exosomes in regenerative medicine and sports health, and also provides a new idea for revealing the close relationship between the heart and other organ systems.

## 1. Introduction

The human body is a complex and closely organized organism that is composed of various tissues and cells that constantly transmit information to maintain normal physiological activities of the body. There are many types of signals that transmit messages between cells, such as chemical messages secreted by cells, direct contact between the membranes of two adjacent cells, and the formation of channels between some plant cells. The internal environment of the human body contains various enzymes, and many biological macromolecules that are released into the peripheral circulation will be degraded and inactivated by corresponding enzymes, thus losing their original functions. In this environment, a complex extracellular vesicle (EV) system is developed for long-distance contact and information transmission between cells. Therefore, compared with the direct release of cytokines, such as proteins, the biggest advantage of EVs for information transfer lies in its strong vesicle membrane, which can protect the contents from the degradation of enzymes in the peripheral circulation or the clearance of the mononuclear phagocytic system [1]. In addition, through EVs system, a variety of information molecules will be transmitted together, so that the transmission efficiency of biological information is greatly increased. In fact, the existence of EVs has been known by researchers for a long period of time, and the mystery of their function and structure has been gradually identified with the in-depth research; thus, the meaning of EVs has also been constantly enriched and developed. The latest interpretation comes from a 2018 position statement (MISEV2018) by The International Society for Extracellular Vesicles (ISEV). In this context, EVs are generically defined as “particles naturally released from the cell”, which are surrounded by lipid bilayer and cannot be replicated; they are vesicles that contain no functional nucleus [2]. Although the naming method of EVs remains to be further improved, EVs are categorized into exosomes (EXOs), microvesicles (MVs), and apoptotic bodies (ABs) according to cell sources and particle size [3]. EVs are now considered another communication mechanism between cells, in addition to hormone transmission. It can carry a variety of biomolecules, such as proteins, nucleic acids, and lipids, and can participate in the material exchange and information exchange between various cells of the body [4].

EXOs originate from intracellular bodies called “multivesicular bodies” (MVBs) and are the smallest particles in EVs, ranging in diameter from 40 nm to 100 nm. EXOs are membrane vesicles that are released into the extracellular environment via exocytosis after MVBs fuse with the cell membrane [5]. Existing studies have shown that EXOs have a density of 1.13–1.19 kg/L, a circular or cup-shaped structure under electron microscopy, and can be precipitated at 100,000× *g* centrifugation [6]. When EXOs were first discovered, they were considered by scholars at the time as a way to expel unwanted components from cells [7]. As the research progressed, EXOs has been confirmed to be able to pass receptor–ligand interaction, exosomal internalization, and direct membrane fusion. These three methods deliver biomolecules to target cells and mediate cell-to-cell communication [8]. EXO vesicles are rich in lipids, proteins, and various types of nucleic acids, such as DNA, mRNA, microRNA (miRNA), and long noncoding RNA (IncRNA). Due to the small size of EXOs, there are usually no organelles, such as ribosomes and mitochondria. The components of different EXOs vary depending on their origin cells, recipient cells, and their functions. In recent years, with the deepening of research, EXOs mediated intercellular communication is deeply involved in the regulation of epigenetic inheritance. Non-coding RNAs contained in EXOs can induce changes in genetic traits by regulating gene expression in target cells. EXOs, as a new research direction and tool, have been applied to the study of epigenetics. For example, Padmasekar et al. [9] used EXOs as tools to study the epigenetic adaptation mechanism of high-altitude humans, and Shrivastava et al. [10] used EXOs as delivery vectors to achieve stable epigenetic inhibition of HIV-1 virus. In addition, EXOs are an important part of the body’s immune response mechanism. It can be involved in intercellular communication in the immune system, including antigen presentation, natural killer cell and T cell activation/polarization, immunosuppression, and various anti-inflammatory processes [11]. Moreover, EXOs also play an important role in the occurrence and development mechanism of various autoimmune diseases, including cancer [12]. It has been found that EXOs carry a variety of biomolecules, such as proteins, nucleic acids, and lipids, which can participate in the material exchange and information exchange between various cells of the body. EXOs derived from different types and cells in different microenvironments carry different types and quantities of contents. In addition, the contents of EXOs that are released by the same cell can vary greatly under different physiological or pathological conditions. Therefore, EXOs can be used as specific biomarkers for some diseases. For example, a reduction in circulating microRNAs in patients with chronic kidney disease was found to be associated with a reduction in the estimated glomerular filtration rate; therefore, Borges et al. [13] proposed that the total number of microRNAs and certain specific microRNAs could be used as biomarkers for uremia. In addition, EXOs have great application prospects in the targeted therapy of many diseases.

In this paper, we review the EXOs derived from heart tissue cells—including cardiomyocytes, fibroblasts, endothelial cells, cardiac resident stem cells which including cardiac progenitor cells (CPCs) and cardiac globule derived cells (CDCs), cardiac adipose-derived stem cells (ADSCs) and cardiac mesenchymal stem cells (MSCs). Additionally, we introduce the components and functions of cardiogenic EXOs, and the factors affecting the secretion and composition of cardiogenic EXOs. Additionally, we deeply discuss the influence of cardiogenic EXOs on cardiac function and the role of cardiogenic EXOs in regulating of distant organ systems. It provides a reference for the development and application of EXOs in regenerative medicine and sports health, and also gives a new sight for revealing the close relationship between heart and organ systems.

## 2. Structure and Function of Cardiogenic EXOs

The heart is a complex assembly composed of cardiomyocytes, Purkinje fibers, and other types of cells. It provides power for blood circulation throughout the body and is the core organ of the human circulation system. EXOs can be released by a variety of cells that constitute the heart [14,15]. Sanjiv et al. [16] first have proved that cardiac myocytes could release EXOs in 2007. In subsequent studies, researchers gradually found that these cell populations could complete information communication in the heart through autocrine or paracrine EXOs, and release EXO secretions into the peripheral circulation, thus mediating remote communication between the heart and other tissues and organs [17]. Cardiomyocytes are known to secrete cytokines that mediate cell-to-cell communication, and EXOs are additional cell-to-cell communication mechanisms. The two systems are not independent, but can influence each other. For example, the treatment of growth factors can induce changes in the transcription content of EXOs released by cardiomyocytes [18]. In turn, EXOs can also affect the endocrine function of their target cells and regulate the process of secretion and release of cytokines. EXO vesicles are biomolecules that convey specific information. EXO vesicles contain special molecules that are specific to their source cells and recipient cells. These molecules allow different EXOs to carry specific information, regulate a variety of complex physiological processes, and provide a variety of topics for further scientific research. For example, some scholars have found that the mRNA carried by EXOs in mouse cells is ingested by human mast cells in vitro; and can then be translated into human mast cells to produce new mouse proteins. Therefore, EXO-mediated mRNA and miRNA transfer is a new mechanism for genetic material exchange between cells [19]. In addition, Quesenberry et al. [20] have reported that EXOs from the lung or liver can enter the bone marrow cells in vitro and induce the expression of original lung or liver tissue specific proteins. Although EXOs are highly specific as a cellular product, there are some molecular components that are common in all EXOs. For example, markers of four transmembrane proteins (CD9, CD63, CD81), Annexins and Flotillin, heat shock proteins (HSP70, HSP90), and endosomal sorting complex required for transport (ESCRT) pathways, including LAMP1 and TSG101 [21].

EXOs derived from heart cells have their specific composition and function. Proteomic analysis of EXOs derived from cardiac myocytes by Malik et al. [22] showed that EXOs derived from cardiomyocytes contain more than 40 proteins not found in other EXOs, which are rich in myofibrin and mitochondrial proteins, including tropomyosin, myosin, cardiac myosin binding protein-C (CMYBP-C), and valosin protein (VCP). In addition, other proteins rich in cardiogenic EXOs include HSP20, HSP60, HSP70, tumor necrosis factor-α (TNF-α), interleukin-6 (IL-6), and glucose transporter1 (GLUT1), GLUT4, etc. [23,24]. Lipids in EXOs have not been the focus of research in the field of EXOs. Research confirms that the lipids rich in EXOs mainly include phingolipids, cholesterol, phosphatidylserine, Phosphatidylinositol-3-phosphate, and bis-Monoacylglycero-phosphate (BMP), etc. [25]. Some lipid molecules in EXOs also have their unique functions. For example, eicosanoids derived from different EXOs can remotely regulate immune responses and be used as biomarkers for some diseases [26]. However, there are no high-quality studies on lipid profiles in cardiogenic EXOs. Waldenstrom et al. [27] have shown that cardiac EXOs contain more than 1500 different mRNA transcripts and 340 different DNA sequences. The transfer of cardiac EXOs to cultured fibroblasts resulted in changes in the properties of the transcriptome-induced 333 gene expressions, including 175 upregulations and 158 downregulations, thus suggesting that cardiac EXOs carry bioactive factors that affect gene expression [27]. Due to the high demand for energy, cardiomyocytes contain a large number of mitochondria and dense protein turnover; therefore, EXOs contain a large number of ribosomal mRNAs. In addition, Genneback et al. [18] have found that more than 200 common transcripts (mature mRNA for encoding proteins) can stably exist in cardiac EXOs, among which ribosomal, cytoplasmic, and mitochondrial transcripts related to energy supply were especially abundant. MiRNAs are short non-coding RNAs that regulate gene expression at the mRNA level. After miRNAs are delivered to the receptor, it regulates the expression of target genes in the physiological and pathological process. Therefore, miRNAs can be said to be an important molecule responsible for the information transfer function of EXOs. Genomic-based studies have confirmed the specificity of miRNAs expression in different human tissues [28]. At present, several studies have reported that EXOs released by different cells have different miRNAs enrichment profiles [29,30]. The composition of miRNAs in EXOs of different heart cells and under different activation states is different. So far, a variety of miRNAs that are often enriched in EXOs produced by heart cells have been found, such as miR-320, miR-30a and miR-29b in cardiomyocyte EXOs; miR-451, miR-144, miR-21, miR-146a in CPCs and CDCs EXOs; miR-21 in fibroblast EXOs; and miR-214 and miR-146a in endothelial cell EXOs [1,24,31,32]. In addition, some studies have also reported miRNAs that specifically exist in heart tissue cells, such as miR-1 [33], miR-208a [34], and miR-195 [35]. In summary, although several studies have provided a general overview of the types and functions of proteins, lipids, and miRNAs commonly found in cardiogenic EXOs. However, no systematic and comprehensive studies about profiles of proteins or miRNAs have been reported by omics deep sequencing. However, fortunately, thanks to the efforts of scholars around the world, several available online databases have been created and improved, Vesiclepedia (http://microvesicles.org, accessed on 10 April 2022), ExoCarta (http://www.exocarta.org, accessed on 10 April 2022). As more and more research data are contributed, the true nature of the molecular composition and lineage profile of cardiogenic EXOs will be gradually revealed.

EXOs have been recognized as a carrier of bioactive molecules from source cells, and can regulate the physiological processes of recipient cells, and play an important role in mediating intercellular signal transduction. Therefore, some scholars call it “extracellular organelle”. Cardiogenic EXOs also have different composition characteristics and missions from EXOs that are derived from other tissues and organs due to the unique role of the heart in the body.

## 3. Cardiogenic EXOs Regulate the Cardiovascular System

The various types of cells that make up the heart can release EXOs under various physiological and pathological conditions, regulate cell function, and perform many related physiological processes. Under pathological conditions, heart cells are still the main target of cardiogenic EXOs, and the most important function of EXOs is to cope with pathological injury, protect the heart and promote the regeneration of tissue cells. The types of cardiogenic EXOs that perform this process are very rich in number. Due to the large number and variety of bioinformatics molecules contained in EXOs that are derived from different cells, they provide rich materials for related research. In recent years, significant results have been achieved in this field. The production of these EXOs is an important physiological response of the body to respond to adverse stimuli and protect important organs under emergency conditions, as well as being an important carrier for mediating pathological processes in various disease states. Additionally, EXOs play an important role in the exchange of information between various cardiac cells and the regulation of cell functions. According to current studies, the miRNAs in cardiogenic EXOs may be the key molecules that explain the role of EXOs.

### 3.1. Positive Regulation of Cardiogenic EXOs on the Cardiovascular System

#### 3.1.1. Anti-Fibrosis Effect

Cardiac fibrosis is a repair response after tissue injury to ensure the relative integrity of tissues and organs. The proliferation of fibrous connective tissue repaired the defect but did not have the structure and function of the original organ parenchymal cells. If this repair response is overactive and out of control, it will cause fibrosis and lead to the decline of organ function. Therefore, in the process of cardiac tissue repair, fibrosis should be controlled to preserve the normal level of cardiac function as much as possible. Studies have shown that many types of heart cells can secrete EXOs to inhibit cardiac fibrosis. EXOs derived from cardiomyocyte play a protective role in cardiac function injury caused by a variety of diseases. For example, HSP20 rich in cardiogenic EXOs can alleviate fibrosis and promote angiogenesis in diabetic mice through Tsg101 pathway [36]. In addition, in a patient population with Duchenne muscular dystrophy (DMD), cardiogenic EXOs also showed significant anti-fibrotic activity via the ERK1/2-P38-MAPK signaling pathway. The cardioprotective effect is transmitted in the form of delaying cardiac fibrosis and cardiomyopathy caused by DMD [37]. Yang et al. [38] used EXOs derived from cardiac terminal granulocytes in their experiment to treat rats induced with myocardial infarction, and the results showed less cardiac fibrosis, decreased collagen deposition, increased angiogenesis, and improved cardiac function. These results demonstrate that cardiac terminal granulocyte-derived EXOs can also reduce the fibrosis process during cardiac remodeling and promote cardiac repair (Figure 1).

#### 3.1.2. Protect Myocardial Cells: Anti-Apoptosis Effect

Chen et al. [39] found that EXOs isolated from cardiac progenitor cells can be used in the cardioprotective treatment of cardiovascular diseases, such as myocardial cell apoptosis and death caused by acute ischemia/reperfusion injury; additionally, EXOs can inhibit oxidative stress-induced myocardial cell apoptosis in vivo and in vitro by inhibiting the activation of caspase 3/7. Oxidative stress can induce cardiac progenitor cells to secrete a large number of EXOs. Xiao et al. [40] reported that under oxidative stress, the miR-21 content in EXOs derived from cardiac progenitor cells was significantly upregulated. EXO miR-21 plays a key role in blocking the apoptosis pathway of H9C2 cardiomyocytes by targeting the downregulated expression of the programmed cell death 4 (PDCD4) gene to protect cardiomyocytes from oxidative stress injury in an ischemic myocardial injury model. In addition, miR-210 enriched in cardiac progenitor cell derived EXOs inhibits cardiac apoptosis by down-regulating the expression of its target proteins ephrin A3 and PTP1b [41]. Normally, adipocytes are scarce in heart, but adipose-derived exosomes (ADSCs-EXOs) can protect heart tissue in a variety of ways. Luo et al. [42] reported that ADSCs-EXOs can reduce the myocardial injury area of myocardial infarction, especially after treatment with miR-126-rich EXOs. Thus, the level of cardiac fibrosis and the expression of inflammatory cytokines are reduced and there is significant promotion of vascular formation in the infarct area of myocardial infarction rats, thus ultimately protecting myocardial cells (Figure 1).

#### 3.1.3. Promote Angiogenesis Effect

After tissue and organ injury, the generation of blood vessels and blood flow reperfusion are essential program for the repair and regeneration of any tissue. For the heart, especially after the occurrence of many ischemic diseases, myocardial tissue will appear to be at different degrees of necrosis and apoptosis, resulting in serious cardiac dysfunction. It is particularly important to restore the blood supply to the necrotic area and complete the repair and regeneration of the damaged tissue as soon as possible. Many studies have confirmed that various types of cells in heart tissue can secrete and produce EXOs that can promote angiogenesis, but their biological composition and mediated physiological mechanism are different. Van Balkom et al. [43] have found that endothelium-derived EXOs can stimulate the migration of recipient cells and promote angiogenesis, including miR-214, which is a miRNA that controls endothelial cell function and angiogenesis, and plays a leading role in EXO-mediated intercellular signaling. EXOs containing miR-214 can inhibit the expression of mutated ataxic telangiectasia in recipient cells, thereby delaying aging and allowing for vascular formation [43]. In addition to endothelial cells, cardiac progenitor cells can also play the same role. For example, a large amount of miR-132 is enriched in EXOs isolated from cardiac progenitor cells, which stimulates and enhances endothelial vascular formation via the targeted downregulation of RasGAP-P120 in endothelial cells [41]. It is noteworthy that the effect of cardiac progenitor EXOs is further enhanced under hypoxia, and counters hypoxia. In addition, recent studies have shown that miR-21-5p from cardiac terminal granulocyte EXOs inhibits the apoptosis of cardiac microvascular endothelial cells by targeting the silencing of the cell death inducing p53 target 1 (Cdip1) gene under hypoxia and ischemia conditions. Therefore, miR-21-5p can reduce the size of cardiac infarction after myocardial infarction, improve cardiac function, and increase cardiac angiogenesis [44]. The production and secretion of vascular endothelial growth factor (VEGF) are closely related to the formation of blood vessels. Tseliou et al. [45] revealed the discovery of a new mechanism for amplifying EXOs bioactivity. They found that the treatment of EXOs secreted by CDCs changed the genetic phenotype of fibroblasts, transformed inert fibroblasts into cells with therapeutic activity, made them secrete A large amount of VEGF, and changed the miRNA profile of fibroblast EXOs, which ultimately played a good role in promoting angiogenesis and cardiac protection [45]. MiR-126 in EXOs released by endothelial progenitor cells can also promote angiogenesis by up-regulating the expression of VEGF [46,47]. In addition to targeting VGEF, Zhang et al. [48] also demonstrated that HSP20 contained in cardiomyocyte EXOs can promote vascular endothelial cell proliferation and angiogenesis by activating endothelial growth factor receptor 2 (VEGFR2). Although cardiogenic EXOs have positive effects on endothelial cell functions and angiogenesis in most cases, they may also have negative effects under the influence of some diseases and other factors. For example, Wang et al. observed that the EXOs secreted by cardiomyocytes of rats with type 2 diabetes contain a large amount of miR-320, which is an antiangiogenic miRNA, and these EXOs can be absorbed by cardiac endothelial cells. Subsequently, miR-320 targets the angiogenesis related genes of cardiac endothelial cells—insulin-like growth factor-1 (IGF-1), HSP20, and v-ets erythroblastosis virus E26 oncogene homolog 2 (Ets2), ultimately leading to the inhibition of endothelial cell proliferation, migration, and tubular formation [49,50] (Figure 1).

### 3.2. Negative Regulation of Cardiogenic EXOs on the Cardiovascular System

In addition to these positive effects above all, cardiogenic EXOs also mediate some pathological regulatory effects in many pathological processes.

The continuous reaction of human heart tissue to many physiological or pathological stimuli, such as sports, pregnancy, high blood pressure, and ischemia, will activate the mast cell reaction of the heart [51], and subsequently cause an increase in the volume of myocardial cells, cardiac fibroblast cell proliferation, the extracellular matrix, and final myocardial hypertrophy. Cardiac hypertrophy is an adaptive compensatory response that improves cardiac reserve and cardiac output. Generally, cardiac hypertrophy caused by physiological factors is reversible, whereas cardiac hypertrophy caused by various cardiovascular diseases is mostly in the form of pathological remodeling, which tends to expand the heart cavity and eventually leads to heart failure and even death. The control and intervention of pathological myocardial hypertrophy has always been a focus of relevant research. During the occurrence of myocardial hypertrophy, EXOs play an irreplaceable role in mediating intercellular communication between cardiomyocytes and other cells (such as fibroblasts and endothelial cells). Therefore, the study of cardiogenic EXOs can also provide new insights into the pathogenesis of myocardial hypertrophy. EXOs can regulate epigenetic modifications during cardiac repair. For example, Morelli et al. [35] confirmed that myocardial cell-specific miR-195 is upregulated in myocardial cells after ischemic injury and transferred to cardiac fibroblasts in the form of exosomal cargo, thus mediating the activation and phenotypic transformation of cardiac fibroblasts. In addition, Bang et al. [52] found that EXOs derived from cardiac fibroblasts mediate myocardial hypertrophy, and contain a high concentration of miR-21-3p, which acts as an intercellular signaling molecule between cardiac fibroblasts and cardiomyocytes during myocardial hypertrophy. During this process, miR-21-3p is transferred into cardiomyocytes to function and downregulate the expression of sorbin and SH3 domain-containing protein 2 (SORBS2) and PDZ and LIM domain 5 (PDLIM5), thus resulting in a large number of cardiomyocyte volume increases [52]. In addition, recent studies have also shown that miR-200a in adipocyte derived EXOs can be delivered into cardiomyocytes, thus leading to decreased expression of TSC1 and subsequently leading to increased activation of mammalian target of rapamycin (mTOR). Ultimately, myocardial cell hypertrophy occurs [53] (Figure 1).

It is important to note that physiological hypertrophy of the heart muscle that occurs during pregnancy can develop into a pathological nature known as peripartum cardiomyopathy (PPCM), which is a serious and potentially life-threatening heart disease. PPCM is characterized by life-threatening sudden heart failure in the last month of pregnancy and/or the first few months after delivery, but its etiology is not yet fully understood [54]. Halkein et al. [55] found that cardiac fibroblasts and endothelial cells can release miR-146a-rich EXOs, thus allowing for the transport of miR-146a to cardiomyocytes. Increased miR-146a content in cardiomyocytes leads to decreased expression of human epidermal growth factor receptor 4 (Erbb4), Notch1 and interleukin 1 receptor associated kinase 1 (Irak1) genes in cardiomyocytes. This will result in decreased metabolic activity and impaired systolic function of myocardial cells [55]. In addition, researchers noted that plasma levels of EXO miR-146a were significantly higher in patients with acute PPCM than in healthy postpartum controls and patients with dilated cardiomyopathy [55]. Therefore, cardiogenic EXOs miR-146a may be a key factor in the pathogenesis of PPCM, and can be used as a new diagnostic marker and a new therapeutic target (Figure 1).

In addition to cardiac hypertrophy and remodeling, angiogenic EXOs have also been reported to play a role in arrhythmia. For example, during ischemia, EXOs secreted by cardiomyocytes are rich in miR-1 and miR-133, which can target Ca^2+^/calmodulin-dependent protein kinase 2 and affect the action potential and cardiac conduction system, thus triggering arrhythmias [56] (Figure 1).

Zhan et al. [57] have found that there are different levels of oxidized low-density lipoprotein (OX-LDL) and homocysteine (Hcy). Hcy can induce a large increase in the content of heat shock protein 70 (HSP70) in EXOs that are released by rat endothelial cells, and HSP70 alone activates monocytes, thus leading to monocyte adhesion to endothelial cells and vascular endothelial dysfunction [57]. This also demonstrates that endothelial cell-derived EXOs play an important role in atherosclerotic plaque formation (Figure 1).

In conclusion, as carriers of miRNAs, proteins, and other biomolecules, EXOs play a role in transmitting disease information in the pathological process of various cardiovascular diseases. If we specifically block the production of EXOs in the progression of disease, it may produce therapeutic effects. Therefore, many physiological and pathological effects of EXOs that are derived in the disease state need to be further clarified to benefit human beings.

## 4. Cross-Organ Regulation of Motor System by Cardiogenic EXOs

In a narrow sense, the motor system of the human body is composed of three organs: bone, bone connection, and muscle. During movement, the motor system can play a role quickly and accurately without the close cooperation and mobilization of other organs. Among them, nerves and the heart are most closely related to motor system. Nerves can transmit nerve impulses and play a dominant role in muscle contraction. The heart supplies blood to skeletal muscle during strenuous exercise and the level of an individual’s heart function directly determines his exercise ability. Many tissues and organs in the human body, such as skeletal muscle [58] and the heart [59], produce unique proteins under various conditions and release them into the circulating blood, exerting profound influence on the metabolism, quality regulation, and immune function of distant tissues. These proteins are called cytokines [60]. In recent years, a large number of studies in the field of EXOs have focused on the miRNAs contained in EXOs, and it has been found that the miRNAs contained in EXOs, such as proteins and other substances, are one of the important specific undertakers of the regulation of EXOs. Similar to cytokines, miRNAs can also be released into the peripheral circulation by some tissues, and can be specifically taken up by distant target cells to produce corresponding effects [61]. Therefore, although the methods for miRNAs to enter the circulation and be used by cells in other organ systems have not been thoroughly explored, the long-distance communication between cells using EXOs as carriers and miRNAs as information regulatory substances must play an irreplaceable role in various cross-organ regulation methods of the body.

### 4.1. Neural Regulation by Cardiogenic EXOs

All kinds of movements produced by the human body are completed under the regulation of the nervous system, so we also include the brain and nerves in the motor system in this paper. The cerebral cortex is the highest nerve center, which is divided into multiple functional areas, and the control of movement is extremely complex. At present, there have been many studies on the regulation of neural function by EXOs, and the sources of these EXOs are also very rich, including mesenchymal stem cell derived EXOs (MSC-EXOs), adipocyte derived EXOs, neuron derived EXOs, and endothelial cell derived EXOs. EXOs released by many cells have been proved to have various effects on the nervous system. For example, MSC-EXOs can transfer many genetic materials, neurotrophic factors and proteins, to axons, restore microenvironment homeostasis, regulate axon regeneration, and thus promote the repair of peripheral nerve injury [62]. EXOs secreted by differentiated and mature C2C12 myotube cells can significantly promote the growth and differentiation of neuronal processes and improve the generative capacity of neurons [63].

For EXOs derived from heart cells, the relevant research results are relatively limited. The heart contains a small amount of adipose tissue located in the epicardium, but, in the pathological state, there is a large accumulation of adipose tissue, which contains visceral fat cells and ADSCs. ADSCs are stem cells with multi-differentiation potential, capable of self-replication and cloning, and secreting and releasing EXOs with multiple regulatory functions. Many studies have confirmed that ADSCs-EXOs have significant therapeutic effects in promoting neural regeneration. Chen et al. [64] found that after internalized by Schwann cells, ADSCs-EXOs significantly promoted proliferation, migration, myelination, and secretion of neurotrophic factor of Schwann cells by up-regulating corresponding genes in vitro, and played a role in promoting sciatic nerve regeneration. In addition, Bucan et al. [65] confirmed that ADSCs-EXOs contains brain-derived neurotrophic factor (BDNF), insulin-like growth factor-1 (IGF-1), nerve growth factor (NGF), glial cell-line-derived neurotrophic factor (GDNF), etc. They can stimulate the proliferation of Schwan cells and promote the growth of axons. As a multifunctional protein, pigment epithelial-derived factor (PEDF) has anti-inflammatory and antioxidant properties and can protect cultured cortical neurons by inhibiting oxidative stress and cell apoptosis. Huang et al. [66] modified ADSCs-EXOs with PEDF and found that it activated autophagy and strongly inhibited neuronal apoptosis, thus improving the prognosis of brain injury. In addition to ADSCs, vascular endothelial cells can also produce EXOs, which play a similar role. Xiao et al. [67] demonstrated that endothelium-derived EXOs promoted the expression of B cell lymphoma-2 (BCL2) in SH-SY5Y neurons and inhibited the expression of BCL2-associated X (Bax) and caspase-3. Therefore, it can directly protect nerve cells to reduce ischemia/reperfusion injury after cerebrovascular disease by promoting cell growth and migration and inhibiting apoptosis. In a word, the current results show that EXOs from cardiac tissue cells can easily penetrate the blood–brain barrier and activate some biological functions of target cells, further revealing their potential as regenerative therapy tools after nervous system injury (Figure 2A).

In addition, studies have found that the cognitive function of the brain is also affected by the regulation of cardiogenic EXOs, and some heart diseases can even lead to the deterioration of the cognitive function of the brain, which is the concept of “cardiogenic dementia”. MiR-1 is a heart-specific miRNA that is expressed in abundance only in the heart and is approximately 100 times lower in the brain than in the heart. Sun et al. [33] found that myocardial infarction can cause cardiac cells to secrete a large amount of miR-1-rich EXOs, which are enriched in the blood circulation and in the hippocampus, thereby causing damage to neuronal microtubules. In addition, heart-derived miR-1 can target the TPPP/p25 gene and reduce the expression of TPPP/p25 protein in the hippocampus. Similarly, Duan et al. [68] reported that overexpression of miR-1 in the heart may lead to an increase in the level of miR-1 in the hippocampus and a decrease in synaptic exocytosis, the molecular mechanism of which involves miR-1 targeting the 3′UTR of the Snap25 gene and inhibiting the expression of SNAP-25 protein after transcription. These findings also shed light on some of the molecular mechanisms underlying heart–brain communication under pathological conditions (Figure 2A).

### 4.2. Skeletal Regulation by Cardiogenic EXOs

The adult skeletal system consists of 206 bones, consisting of periosteum, bone, and marrow, as well as a rich distribution of blood vessels and nerves. Bone tissue consists of four types of cells: osteoprogenitor, osteoblast, osteocyte, and osteoclast. Bones can support and protect the human body. Bone marrow has the function of hematopoietic, and, in addition, bone is an important reservoir of calcium and phosphorus. All types of cells in the human body secrete EXOs, which mediate cell-to-cell communication. The various cells that make up bone tissue are also regulated by signals from other cells, and effective communication between these cells is essential for maintaining bone homeostasis. EXOs from different cells have different effects on bone tissue. It should be noted that current research on EXOs and bone mainly focuses on the cells that make up bone tissue itself. For example, EXOs secreted by bone marrow mesenchymal stem cells (BMSCs), osteoclasts and osteoblasts have been proved to participate in the regulation of bone homeostasis. However, there are few studies on the regulation of bone by EXOs from extra-bone organs, especially the effects of EXOs from the heart on bone tissue.

Vascular endothelial cells are a layer of flat epithelium that lines the inner surfaces of the heart, blood vessels, and lymphatics vessels. Song et al. [69] found that naturally formed endothelium-derived EXOs can inhibit osteoclast activity in vitro and reduce bone resorption in vivo. The possible mechanism for these effects is related to the high concentration of miR-155 in vesicles [69]. Moreover, EXOs from endothelial cells showed better bone targeting ability than EXOs from osteoblasts and BMSCs. This phenomenon also provides a new target for the prevention and treatment of osteoporosis. In addition to endothelial cells, endothelial progenitor cells can also indirectly affect bone tissues by secreting EXOs. An in vitro analysis showed that endothelial progenitor cell-derived EXOs can enhance the proliferation, migration, and angiogenesis of endothelial cells, and miR-126 in its vesicles is the most important information molecule mediating this process [70]. Furthermore, bone regeneration can be accelerated by stimulating angiogenesis [70] (Figure 2B).

In addition to promoting bone growth, development, and regeneration, some EXOs have the opposite regulatory effect. Weilner et al. [71] found that human endothelial cells produce more EXOs during aging, which are absorbed by BMSCs via the blood circulation, and miR-31 contained in EXOs can inhibit the osteogenic differentiation of BMSCs by targeting the downregulation of Frizzled-3. As a new regulatory mechanism, EXOs also play an important role in the pathogenesis and progression of osteoarthritis (OA). Yang et al. [72] demonstrated via in vivo and in vitro experiments that EXOs produced by vascular endothelial cells can reduce the ability of chondrocytes to resist oxidative stress by inhibiting autophagy and P21 expression, thus increasing cell reactive oxygen species (ROS) content and inducing chondrocyte apoptosis. This finding also provides new ideas for the diagnosis and treatment of OA (Figure 2B).

### 4.3. Regulation of Skeletal Muscle by Cardiogenic EXOs

Skeletal muscle is a type of striated muscle attached to bone, and is the motor part of the motor system. Under the control of the nervous system, skeletal muscle contracts and pulls on bone to produce movement. As the tissue/organ with the largest mass of the human body, skeletal muscle not only produces movement, but also maintains body posture, protects internal organs, produces heat, pumps blood vessels, and endocrine functions, which are very important for homeostasis regulation of the body. Some EXOs secreted by heart cells enter the blood and reach skeletal muscle through blood circulation, playing an important role in regulating the function of skeletal muscle.

As mentioned above, the long-distance communication between cells with EXOs as the carrier and miRNA as the information regulatory substance plays an irreplaceable role in various cross-organ regulation methods of the body. Multiple in vitro studies have demonstrated that miRNAs contained in EXOs can be transferred from external mediators to recipient cells, where they affect their biological functions [19]. Recent studies have found that this communication mode plays an important role in the heart–skeletal muscle connection, mediating a variety of physiological and pathological interactions.

In the human body, many miRNAs are only specifically expressed in certain tissues, and some of them are only specifically expressed in striated muscle tissues, known as myomiRs [73]. Currently known myomiRs include miR-1, miR-133a, miR-206 (expressed only in skeletal muscle), miR-208a (expressed only in myocardium), miR-208b, miR-499, and miR-486 [34]. It is well known that skeletal muscle and cardiac muscle are morphologically classified as striated muscle. Most of the above-mentioned myomiRs are specifically expressed in skeletal muscle and cardiac muscle, and there is now ample evidence that miRNAs in the human heart are released into the circulation under various stimuli. Therefore, it is quite possible that myocardial derived miRNAs can be released into circulation and affect skeletal muscle in various physiological and pathological conditions.

Heart failure is the late state of a variety of heart diseases and manifests as severe cardiac dysfunction. There are many causes of heart failure. However, some scholars have found that several miRNAs frequently appear in studies related to heart failure, and they are changed in various heart diseases [74], including miR-1, miR-21, miR-24, miR-29b, miR-133a, miR-199, miR-208, miR-214, and miR-499. It is not difficult to find that most of these miRNAs are contained in the myomiRs sequence. Therefore, this also provides strong evidence for the continuous release of miRNA-rich EXOs from the heart during heart failure, thus mediating communication with skeletal muscle. For example, Heineke et al. [75] confirmed that myostatin released from heart tissue during heart failure can strongly regulate skeletal muscle mass. In fact, the expression of miR-1 and miR-133a has also been confirmed to be decreased during skeletal muscle hypertrophy [76]. Therefore, some scholars have similarly hypothesized that the increase in miR-1 and miR-133a in circulating blood from heart failure can promote skeletal muscle atrophy after the development of cardiovascular disease [34]. In addition, in a study of DMD, cardiogenic cell-derived EXOs containing miR-148a were delivered intravenously to skeletal muscle and promoted skeletal muscle regeneration in mdx mice. This proves that EXOs derived from heart cells can enhance skeletal muscle function and improve skeletal muscle regeneration [77] (Figure 2C).

### 4.4. Regulation of Tendons and Ligaments by Cardiogenic EXOs

Tendons are tough, non-elastic, string-like structures that lie at the ends of skeletal muscle and are made of dense connective tissue. Muscles attach to the surfaces of the periosteum, fascia, and joint capsule by means of tendons, and their main function is to transfer the forces generated by muscles to the bones, thereby producing joint movement. As one of the auxiliary structures of synovial joint, ligament is a bundle of dense connective tissue connecting two adjacent bones, which can strengthen the connection between two bones, increase joint stability and limit excessive joint movement. Tendons and ligaments are not only similar in structure, extracellular matrix content, and biomechanical properties, but also prone to injury during strenuous exercise. Due to their special anatomical structure, most of the current treatment methods have poor prognostic effects.

EXOs have been proved to play an active role in the regeneration and repair of many tissues and organs. Therefore, the application of EXOs in tendon and ligament regeneration therapy has gradually become a new research hotspot in the field of sports medicine. A previous study confirmed that ADSC-EXOs can effectively promote tendon healing after injury, and the molecular mechanism may include ADSC-EXOs being absorbed by tendon stem cells to activate the SMAD2/3 and SMAD1/5/9 pathways, thereby promoting the proliferation, migration, and tendon differentiation of these cells [78]. In addition, ADSC-EXOs can inhibit the activity of nuclear factor kappa-B (NF-κB) and reduce the expression of the proinflammatory cytokines Il1b and matrix metalloproteinase-1 (MMP-1) at the injured site. Therefore, ADSC-EXOs can play a role in attenuating tendon inflammation at the earliest healing stage after acute tendon injury and repair [79]. Therefore, ADSC- EXOs have been used for the repair and treatment of rotator cuff injury in rats and have achieved good results [80]. For other structures, such as the Achilles tendon and ligament, many studies have proved that MSC-EXOs have the therapeutic effect of promoting regenerative healing for them, but the therapeutic effect of EXOs derived from heart cells still needs a large number of studies to provide more strong evidence support. However, it is foreseeable that cardiogenic EXOs will have a broad and bright application potential in tendon and ligament injuries and other injuries where traditional therapies are difficult to achieve satisfactory results (Figure 2D).

## 5. Factors Affecting Cardiogenic EXOs

In general, the heart’s primary function is thought to be to serve as a “pump” to power blood circulation throughout the body. Cardiomyocytes are the main cells of the heart, and they are not considered to be typical secretory cells. However, based on recent studies, it has been found that a variety of cells in the heart, including cardiomyocytes, can secrete EXOs and microvesicles. The release of these biomolecules is affected by a variety of physiological and pathological factors and can be promoted by artificial intervention. In addition, the contents of cardiogenic EXO vesicles are influenced by various stimuli.

### 5.1. Effect of Myocardial Infarction on Cardiogenic EXOs

Acute myocardial infarction (AMI) is a kind of acute myocardial ischemic heart disease. With cardiac coronary atherosclerosis as the main underlying disease, plaque rupture in the coronary artery triggers coagulation reaction induces platelet aggregation, and then thrombosis in the vascular lumen and progresses along with the blood flow, ultimately leading to acute ischemic hypoxia necrosis of cardiomyocytes downstream of the blood vessel [81]. The physiological and pathological mechanisms leading to ischemic injury of heart tissue are diverse, among which metabolic acidosis, Ca^2+^ overload and accumulation of ROS are the core elements caused by hypoxia and insufficient supply of essential nutrients for some cells [82]. A series of physiological and pathological changes induced by AMI can stimulate various heart cells and then affect the EXOs they secrete. These stimuli can regulate the secretion of cardiac EXOs and change the content of various biomolecules in EXOs.

After AMI, damaged cardiomyocytes secrete a large number of EXOs. When cardiomyocytes are in an anoxic environment, EXO release is increased, and the severity and duration of the anoxic environment regulate the release of EXOs from cardiomyocytes. For example, after 2 h in a moderately hypoxic environment, the amount of EXOs released by cardiomyocytes increased nearly twice [16]. This process is the body’s adaptive response to tissue damage, thus transmitting information about heart damage to tissues and cells throughout the body. Several studies have proven that EXOs from cardiomyocytes can regulate cell proliferation, migration, differentiation, survival, and promote angiogenesis when ischemia occurs [18,83]. Yellon et al. [84] have shown that EXOs can carry many proteins related to heart protection, such as phosphatase and tensin homolog (PTEN), epithelial growth factor receptor (EGFR), tumor necrosis factor-α (TNF-α), and NAD(P)H oxidase, etc. Normally cardiomyocytes are thought to be permanent, and unable to regenerate once damaged. Damaged myocardial tissue after AMI is gradually replaced by connective tissue or scar without capillaries. EXOs released by damaged cardiomyocytes are rich in heart-specific miRNAs, which seem to play an important role in the above processes [85]. Montgomery et al. [86] demonstrated that miR-208a contained in EXOs secreted by injured myocardial cells after AMI, which showed a pro-fibrosis effect in the heart. Therefore, these EXOs rich in specific miRNAs play an important role in signaling, protecting the heart, and inducing heart tissue repair.

Effects of AMI on EXOs derived from cardiac fibroblasts. Under normal circumstances, cardiac fibroblasts account for about 60–70% of all cardiac cells, accounting for one-third of the normal volume of the heart, and the collagen fibers secreted by them are an important component of the extracellular matrix [87]. When the heart is ischemic and hypoxic, the secreted activity of cardiac fibroblasts is enhanced, which mediates the communication between cardiac fibroblasts and other cardiac cells and affects the physiological activities of other cells. Previous studies have confirmed that EXOs secreted by neonatal rat fibroblasts are enriched in extracellular matrix and selectively up-regulated the expression level of extracellular matrix related proteins under hypoxia conditions [88]. Cardiac fibroblasts and their EXOs play a variety of roles in coping with AMI induced cardiac tissue damage. For example, Bang et al. [52] found that miRNAs in EXOs secreted by cardiac fibroblasts could mediate myocardial hypertrophy. However, other scholars have reported that EXOs derived from fibroblast have protective effects on myocardium after AMI and during blood reperfusion [89,90].

The effect of AMI on EXOs derived from cardiac endothelial cells. Endothelial cells are the building blocks of arteries. When exposed to stress or injury, endothelial cells secrete a variety of biomolecules, including EXOs, and the composition of EXOs varies under different stimulation conditions. For example, in the case of hypoxia, the expression of functional proteins which involved in extracellular matrix remodeling in vesicles increases [91]. Endothelium-derived EXOs play an important role in mediating the formation of blood vessels after cardiac tissue injury in addition to intercellular communication. Van Balkom et al. [43] demonstrated that endothelial cells release EXOs in large quantities after AMI, in which miR-214 mediates information exchange between endothelial cells and stimulates receptor cell migration and angiogenesis.

Effects of AMI on EXOs derived from cardiac stem cells. Small numbers of cardiac stem cells have long existed in the hearts of adult mammals cardiac stem cells are a kind of pluripotent stem cells that can differentiate into various tissue cells, while CPCs are specialized stem cells with limited differentiation and proliferation ability. CPCs can be derived from cardiac stem cells. The highly regenerative properties of these cells enable them to participate in the repair of damaged myocardium after AMI and contribute to the replacement of senescent myocardium cells [92]. CPCs were originally thought to repair damaged cardiac tissue through the classic mechanism of direct differentiation into cardiac muscle cells. With the development of research, people gradually found that the role of them seems to be completed through an indirect path. In other words, a variety of biological signaling molecules, including EXOs, are released through paracrine to mediate cellular communication, thus regulating the growth and differentiation of surrounding cells and other physiological activities, and finally achieving the effect of heart repair. Under the conditions of in vitro culture, these cells spontaneously grow into spherical cell groups, which can be called cardiac spheres [93]. Barile [94] observed, under electron microscopy, that CPCs and CDCs in mice and humans could produce EXOs. Hypoxia induced by AMI increased the release of cardiac progenitor cell EXOs in mice and changed the contents of various molecules [95]. EXOs released from CPCs of mouse in hypoxic environment showed excellent cardiac protective effects, including promoting angiogenesis, alleviating fibrosis during cardiac repair, and resisting apoptosis.

Of course, not all EXOs released by cardiac cells after AMI have positive effects on the body. There are also many kinds of EXOs after the occurrence of disease can induce cardiomyocyte apoptosis, cell hypertrophy, cardiac fibrosis, and other negative reactions. Therefore, more studies are still needed to promote the positive effect of EXOs after AMI, block the related negative effects, and bring more powerful support for clinical treatment of AMI (Figure 3A).

### 5.2. Effects of Ethanol on Cardiogenic EXOs

Ethanol is a hydroxyl radical scavenger, which is transformed into a variety of free radical metabolites after ingestion, among which 1-hydroxyethyl radical is the most abundant (about 80% of the total amount) [96]. Malik et al. [22] found that the intake of ethanol greatly increased EXOs produced by cardiomyocytes and changed the types and contents of proteins in their vesicles, but did not change the permeability of EXO vesicles. They were treated with antioxidants and showed a significant decrease in EXOs content in the ethanol-treated group, suggesting that ROS-mediated signaling may be involved in stimulating EXOs secretion in cardiomyocytes [22]. In addition, protein denaturation and direct cell damage induced by ethanol intake may also be one of the mechanisms for the above changes in cardiac EXOs. In conclusion, ethanol treatment can cause a large amount of EXOs secretion in heart cells, but the specific mechanism pathway, the relationship between ethanol intake and EXOs production, the specific role of the resulting EXOs in vivo and its final destination still need further research (Figure 3B).

### 5.3. Effects of Exercise on Cardiogenic EXOs

In addition to various pathological conditions, physiological stimulation, such as exercise, also plays an important role in EXOs release from the heart.

The effects of exercise on cardiogenic EXOs include promoting the secretion and release of EXOs by related cells and influencing the contents of various components in vesicles. Brahmer et al. [97] found that aerobic exercise can trigger the secretion and release of various extracellular vesicles of various cell types, including cardiomyocytes, endothelial cells, and platelets, and affect cardiac function. In addition, the content of protein components in extracellular vesicles can be significantly altered by exercise. Previous studies have shown that more than 300 proteins involved in the formation of extracellular vesicles were found to be abundant in the peripheral circulation after exercise, among which several proteins involved in the formation of EXOs were significantly enriched in the circulation [98]. Moderate exercise is known to be beneficial for the functioning of the cardiovascular system. Recent studies have shown that EXOs released by cells after exercise play an important role in this process, and, in fact, miRNAs seem to play a key role. Studies have confirmed that sports induced 12 kinds of miRNAs (miR-1-3p, -208a-3p, -486-5p, -23a-3p, -23b-3p, -451a, -16-5p, 378a-5p, -126-3p, -150-5p, -222-3p, and -186-5p) significantly increased in circulating EXOs [99]. Ma et al. [46] confirmed that exercise enhanced the release of EXOs of endothelial progenitor cells and increased the level of miR-126. They regulate the expression of sprouty-related protein-1 (SPRED1) and up-regulate the expression of VEGF by means of SPRED1/VEGF pathway and has a protective effect on vascular endothelial cells. Chaturvedi et al. [100] used db/db mice in type 2 diabetes mellitus (T2DM) model and found that exercise induced the secretion of cardiac EXOs and significantly increased the expression of anti-fibrosis miRNAs (miR-29b and miR-455) contained in these EXOs. These miRNAs inhibit the expression of matrix metalloproteinase (MMP-9), an enzyme involved in extracellular matrix remodeling, and reduce myocardial fibrosis, a pathological manifestation of T2DM [100]. In addition, long-term physical exercise can also promote the production of EXOs miR-342, which is produced by endothelial cells and delivered to cardiomyocytes, and it can reduce cardiomyocyte apoptosis and protect the heart by inhibiting caspase 9 and cardiac C-Jun N-terminal kinase (JNK2) [101].

In conclusion, appropriate physical exercise is a good stimulation for all type of cells in the heart, and the cells also perform extensive intercellular communication via the medium of EXOs, and make corresponding responses to exercise stimulation. The resulting EXOs have extensive and positive effects on the cardiovascular system (Figure 3C).

### 5.4. Effects of other Biological Factors on Cardiogenic EXOs

Genneback et al. [18] used two growth factors—transforming growth factor-β2 (TGF-β2) and platelet-derived growth factor-BB (PDGF-BB) in vitro culture stimulated HI-1 cardiomyocytes, compared with untreated cardiomyocytes. The final results suggest that stimulation of cardiomyocytes with growth factors appears to alter the transcription content of their secreted EXOs, mediating the signaling of receptor cell proliferation, development, and hypertrophy. In addition, the number of EXOs produced by cardiomyocytes stimulated by growth factors is also more advantageous [18] (Figure 3D).

Blood glucose levels in the human body will also affect the release of EXOs, such as neonatal cardiomyocytes in the absence of glucose, the synthesis and release more EXOs, and, thus, increase vascular endothelial cell proliferation and angiogenesis, also increases the activity of glucose uptake, glycolysis and pyruvic acid synthesis in vascular endothelial cells [102,103] (Figure 3D).

Certain drugs can also affect the secretion of cardiac EXOs. For example, it has been reported that Suxiaojiuxin Pill (SJP) can promote the secretion of EXOs from mouse cardiac MSCs in vitro, and it has been proved that this process is regulated by SJP through GTPase (in which Rab27b plays a major role) dependent pathway [104] (Figure 3D).

## 6. Conclusions

EXOs are soluble cellular mediators secreted by a variety of cells. They transfer mRNA, lipids, siRNA, proteins, miRNA, and other biomolecules to adjacent or remote cells, and play an important role in cell communication and epigenetic regulation. Our findings suggest that EXOs derived from cardiac tissue cells can regulate nerves, muscles, bones, and other tissues and organs across organ distances, which not only by promoting their repair after injury, but also by playing an important role in the pathophysiology pathogenesis and the development of some motor system injuries. EXOs have been widely studied in the field of regenerative medicine because of their good ability to promote tissue repair and their advantages of low risk and good effects compared with traditional stem cell transplantation therapy. In addition, the functions of EXOs in epigenetic, immune response and endocrine fields have been gradually revealed. The application field of EXOs has been further expanded. In recent years, research on cardiogenic EXOs has exploded, the majority of current studies focus on their application in various cardiovascular diseases, including their use as diagnostic biomarkers for early onset of disease, as drug delivery vectors after bioengineering modification, and new therapeutic methods for repair and regeneration after tissue injury. The human body is a closely related whole, and EXOs may be an important undertaker. EXOs released by heart cells travel to all parts of the body with blood circulation and regulate the physiological and pathological activities of various tissues and organs from a distance. Reviewing the current research results, we found that although the topic of “cross-organ regulation of cardiogenic EXOs” is gradually being reported by more and more scholars, there has not been a comprehensive study focusing on motor system. We have been working in the field of exercise science for many years, creatively exploring the effects of physical exercise on the cardiovascular system and the close relationship between the cardiovascular system and the exercise system from the perspective of EXOs. The study of cardiogenic EXOs has a broad prospect in revealing the mechanism of good effects of physical exercise on the body, the close relationship between the heart and other tissues and organs, and can also provide a new target for the treatment of a variety of diseases. However, more research is clearly needed to provide more evidence that cardiogenic EXOs regulate other tissues and organs across organs.

In order to obtain high purity EXOs, they must be isolated from various extracellular vesicles. The operation is usually based on the differences in size, biological source, and content between EXOs and other extracellular vesicles [105]. Based on this, the most commonly used method for the separation and purification of EXOs is ultra-centrifugal method, in addition to ultrafiltration, enzyme linked immunosorbent assay (ELISA), etc. [106]. In the past decade, the development of EXOs isolation technology has made great progress, which provides a powerful boost to the understanding of EXOs utilization. However, the purification process of EXOs still faces many problems, such as high cost, complex process, low yield, and easy destruction of EXOs. So far, there is still no perfect purification process. This also greatly restricts EXOs from being used more widely in the clinic. Therefore, the purification process of EXOs is also the focus of further research.

However, in addition to the thorough research involving EXOs, there is no doubt that EXOs should be emphasized in biological medicine and sports health as an increasingly promising aspect of application. For example, EXOs isolated by biological engineering can carry specific drugs and be used to treat motion systems, the nervous system, and other related diseases. It is believed that with the development of science, the application of EXOs will benefit human health.

## Figures and Tables

**Figure 1 ijms-23-05765-f001:**
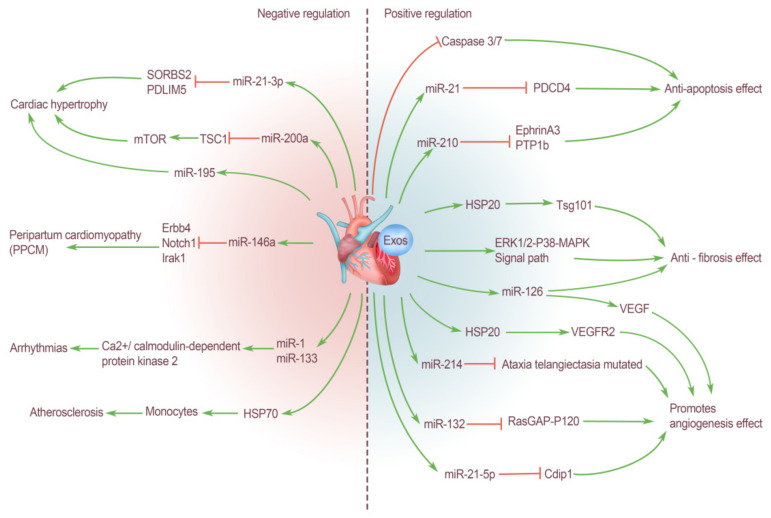
Cardiogenic exosomes regulate the cardiovascular system. Cardiogenic EXOs regulate the cardiovascular system in both positive and negative ways. Positive effect: after cardiovascular system injury, exosomes secreted from heart cells protect the cardiovascular system and promotes repair and regeneration. Cardiogenic exosomes can inhibit myocardial cell apoptosis by inhibiting the activation of caspase 3/7. Cardiogenic EXOs miR-21 blocks the apoptosis pathway by targeting down the expression of PDCD4 gene to protect cardiomyocytes. Cardiogenic exosomes miR-210 inhibits cardiac apoptosis by down-regulating the expression of its target proteins ephrin A3 and PTP1b. Cardiogenic exosomes HSP20 can alleviate fibrosis and promote angiogenesis through the Tsg101 pathway. Cardiogenic exosomes also showed significant anti-fibrotic activity through the ERK1/2-P38-MAPK signaling pathway. MiR-126-rich cardiogenic exosomes can reduce the level of cardiac fibrosis and significantly promote vascular formation. Cardiogenic exosomes miR-214, miR-132, and miR-21-5p can inhibit/silence the expression of mutated ataxic telangiectasia, RasGAP-P120 and Cdip1 genes, respectively, and miR-126 in exosomes released by endothelial progenitor cells can also promote angiogenesis by up-regulating the expression of VEGF, thus allowing vascular formation and increasing cardiac angiogenesis. In addition, HSP20 contained in cardiomyocyte exosomes can promote vascular endothelial cell proliferation and angiogenesis by activating VEGFR2. Negative effects: miR-21-3p and miR-200a in cardiogenic exosomes inhibit the expression of SORBS2, PDLIM5, and TSC1 (and this subsequently leads to increased mTOR activation) myocardial cell-specific miR-195 is upregulated in myocardial cells and transferred to cardiac fibroblasts in the form of exosomal cargo, respectively, leading to cardiac hypertrophy. MiR-146a in cardiogenic exosomes down-regulates the expression of Erbb4, Notch1, and Irak1, which plays an important role in the occurrence of PPCM. During ischemia, cardiogenic exosomes are rich in miR-1 and miR-133, which can target Ca^2+^/calmodulin-dependent protein kinase 2 and affect action potential and cardiac conduction system, thus triggering arrhythmias. Cardiogenic exosomes HSP70 can activate monocytes and play an important role in atherosclerosis.

**Figure 2 ijms-23-05765-f002:**
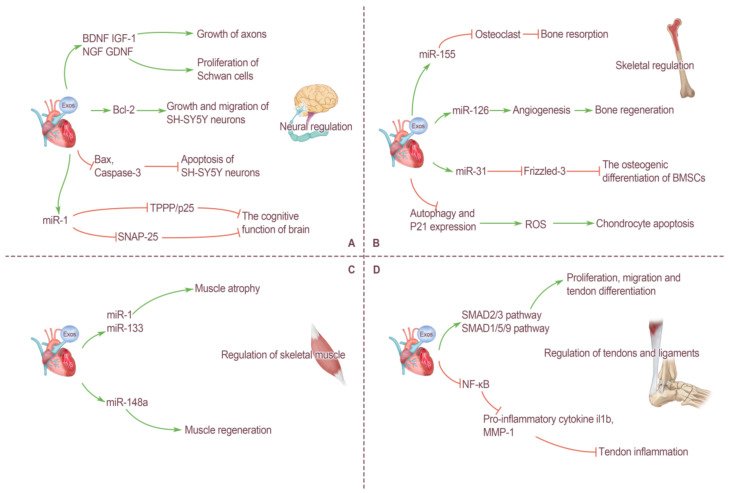
Cross-organ regulation of motor system by cardiogenic exosomes. (**A**) Neural regulation by cardiogenic exosomes. Cardiogenic exosomes contain BDNF, IGF-1, NGF, and GDNF. They can stimulate the proliferation of Schwan cells and promote the growth of axons. Cardiogenic exosomes promote Bcl-2 expression in SH-SY5Y neurons, thereby promoting growth and migration of SH-SY5Y neurons. At the same time, cardiogenic exosomes inhibit the expression of Bax and caspase-3 in SH-SY5Y neurons, thereby inhibiting their apoptosis. Cardiogenic exosomes miR-1 can reduce the expression of TPPP/p25 protein and SNAP-25 protein which, in turn, impairs the cognitive function of the brain. (**B**) Skeletal regulation by cardiogenic exosomes. Cardiogenic exosomes miR-155 can inhibit osteoclast activity and reduce bone resorption. Cardiogenic exosomes miR-126 can enhance the angiogenesis and promote bone regeneration. Cardiogenic exosomes miR-31 act on mesenchymal stem cells, therefore, osteoblast differentiation was inhibited by inhibiting Frizzled-3 expression. Cardiogenic exosomes reduce the ability of chondrocytes to resist oxidative stress by inhibiting autophagy and P21 expression, thus increasing cell ROS content and inducing chondrocyte apoptosis. (**C**) Regulation of skeletal muscle by cardiogenic exosomes. Cardiogenic exosomes miR-1 and miR-133 can promote muscle atrophy. Cardiogenic exosomes miR-148a can promote muscle regeneration. (**D**) Regulation of tendons and ligaments by cardiogenic exosomes. Cardiogenic exosomes can target and effectively activate the SMAD2/3 and SMAD1/5/9 signaling pathways, thereby promoting the proliferation, migration, and tendon differentiation of these cells. Cardiogenic exosomes can inhibit the activity of NF-κB and reduce the expression of pro-inflammatory cytokine IL1b and MMP-1 and play a role in attenuating tendon inflammation.

**Figure 3 ijms-23-05765-f003:**
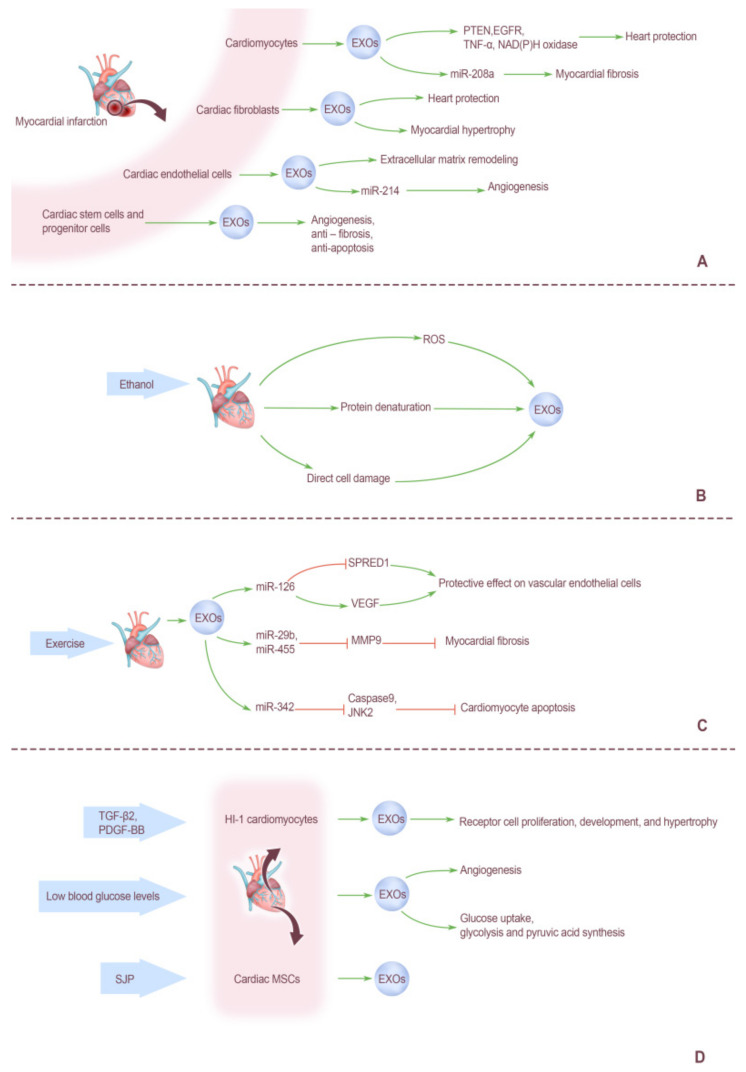
Factors affecting cardiogenic exosomes. (**A**) Effect of myocardial infarction on cardiogenic exosomes. Myocardial infarction increases the secretion of various cell exosomes that make up the heart. PTEN, EGFR, TNF-α, and NAD(P)H oxidase in the secretes of cardiomyocytes had heart protection effects, and miR-208a in them promoted myocardial fibrosis. Exosomes from cardiac fibroblasts can protect the heart and promote cardiac hypertrophy. Exosomes from cardiac endothelial cells promote extracellular matrix remodeling, and miR-214 promotes angiogenesis. Cardiac stem cells and progenitor cells can promote angiogenesis, inhibit fibrosis, and inhibit apoptosis. (**B**) Effects of ethanol on cardiogenic exosomes. The intake of ethanol greatly increased exosomes produced by cardiomyocytes and changed the types and contents of proteins in their vesicles. ROS-mediated signaling, protein denaturation and direct cell damage induced by ethanol intake may be the mechanisms for the above changes in cardiac exosomes. (**C**) Effects of exercise on cardiogenic exosomes. The effects of exercise on cardiogenic exosomes include promoting the secretion and release of exosomes by related cells and influencing the contents of various components in vesicles. MiR-126 can promote the expression of VEGF and inhibit the expression of SPRED1 and has protective effect on vascular endothelial cells. MiR-29b, miR-455 can inhibit the expression of MMP-9, and reduce fibrosis of stem cells. MiR-342 can reduce cardiomyocyte apoptosis and protect the heart by inhibiting caspase 9 and cardiac JNK2. (**D**) Effects of other biological factors on cardiogenic exosomes. Two growth factors: TGF-β2 and PDGF-BB can promote the secretion of HI-1 cardiomyocytes to produce exosomes. Low blood glucose levels can promote the secretion of cardiogenic exosomes, and increase vascular endothelial cell proliferation and angiogenesis, also increases the activity of glucose uptake, glycolysis, and pyruvic acid synthesis. SJP can promote the secretion of exosomes from mouse cardiac mesenchymal stem cells.

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
