# Peer review of "Research Progress on Transorgan Regulation of the Cardiovascular and Motor System through Cardiogenic Exosomes"

_ijms, 2022, doi:10.3390/ijms23105765_

Round 1

Reviewer 1 Report

Comments to authors

The manuscript by Gao et al., “Research Progress of Transorgan Regulation of Cardiovascular and Motor System through Cardiogenic Exosomes” is dedicated review to the important topic of cardiac cell-derived exosomes and their role in the rest of the body. While the idea is great, the execution is rather poor. Reading was very hard, due to the flow of the text. In addition, the word “exosomes” is not so long that it should be abbreviated “EXOs”.  

Many things are just mentioned like facts without any analysis in the larger context. Several times facts are repeated. Several inaccuracies (like: (i) mentioning that motor system is just bones, muscles and bone connections – but what about the brain and nerves? They are mentioned as additional system that helps motor system; (ii) “cardiac fibroblasts account for about 60%-70% of all cardiac cells and constitute the main component of extracellular matrix...” extracellular matrix already means – something outside of cells; components of extracellular matrix are different proteins (like collagen), glycoproteins, proteoglycans etc. ).  

As a positive note, the figures are of high quality.

Main questions:

We know the cardiac cells releases different signalling molecules, including cytokines. How does the exosomes fit into this picture? Are they additional regulators (of what), are they interact with cytokines?

What is the miRNA composition found in exosomes from cardiac cells? How their profile overlaps with the profile of miRNAs from other organs derived exosomes? What specific miRNAs are in cardiac-derived exosomes?

Are there specific miRNA profiles in exosomes isolated from cardiomyocytes, cardiac endothelium etc.? May be start the review with what cell populations are you taking into account when talking about cardiac exosomes otherwise, when reading about MSC-exosomes and adipose tissue-derived exosomes, it is hard to link them to heart exosomes. Also, please, specify - do you review exosomes derived from heart tissue cells or the whole cardiovascular system cells.

What about the other contents of exosomes (proteins and lipids)? What is their profile in cardiac-derived exosomes?

If cells can release proteins, why they need to release exosomes (since they also contain proteins)?

We know that an improvement of angiogenesis is related through secreted VEGF, how cardiac exosomes improves it?

There were too many problematic parts. If you wish to receive detailed analysis (sentence by sentence), please, send the manuscript with line numbers.

Author Response

Response to Reviewer 1 Comments:

Point 1: Extensive editing of English language and style required.

Response 1: Thank you for your important advice. We have asked native English speakers to revise the language of this article. If you are not satisfied sufficiently, we can revise it again. Please find attached the revised proof of this article.

Point 2: The manuscript by Gao et al., “Research Progress of Transorgan Regulation of Cardiovascular and Motor System through Cardiogenic Exosomes” is dedicated review to the important topic of cardiac cell-derived exosomes and their role in the rest of the body. While the idea is great, the execution is rather poor. Reading was very hard, due to the flow of the text. In addition, the word “exosomes” is not so long that it should be abbreviated “EXOs”. 

Response 2: About the flow of the text, we have asked native English speakers to revise the language of this article. And in our article, we've used the word “EXOs” instead of “exosome”(except the abstract part). Please refer to the attached manuscript for details.

Point 3: Many things are just mentioned like facts without any analysis in the larger context. Several times facts are repeated.

Several inaccuracies (like: (i) mentioning that motor system is just bones, muscles and bone connections – but what about the brain and nerves? They are mentioned as additional system that helps motor system; (ii) “cardiac fibroblasts account for about 60%-70% of all cardiac cells and constitute the main component of extracellular matrix...” extracellular matrix already means – something outside of cells; components of extracellular matrix are different proteins (like collagen), glycoproteins, proteoglycans etc. ).

Response 3: Thank you very much for your careful review of the inaccuracies questions rose for us. As for the points you mentioned, all of our authors have carefully reviewed and revised the whole paper, deleted and simplified the repeated facts, and added more analytical comments. 

(i) In the section “4.1 Neural regulation by cardiogenic EXOs”, we have added a clear definition to the category of motor systems. “All kinds of movements produced by the human body are completed under the regulation of the nervous system, so we also include the brain and nerves in the motor system in this paper.” Please see lines 464-466 in section 4.1 for details.

(ii) This inaccuracie is due to my negligence. We've changed the expression. “Under normal circumstances, cardiac fibroblasts account for about 60-70% of all cardiac cells, accounting for one third of the normal volume of the heart, and the collagen fibers secreted by them are an important component of the extracellular matrix.” Please see lines 693-699 in section 5.1 for details.

Point 4: We know the cardiac cells releases different signaling molecules, including cytokines. How does the exosomes fit into this picture? Are they additional regulators (of what), are they interact with cytokines?

Response 4: We believe that cardiomyocytes are known to secrete cytokines that mediate cell-to-cell communication, and EXOs are an additional cell-to-cell communication mechanism. Both are required by cells to communicate with each other, and they are responsible for different physiological functions. But the two systems are not independent, but can influence each other. Please see lines 167-173 in section 2 for details.

Point 5: What is the miRNA composition found in exosomes from cardiac cells? How their profile overlaps with the profile of miRNAs from other organs derived exosomes? What specific miRNAs are in cardiac-derived exosomes?

Response 5: Thank you for your good suggestion. We have already added important content about miRNAs in the article. “At present, several studies have reported that EXOs released by different cells have different miRNAs enrichment profiles[29,30]. The composition of miRNAs in EXOs of different heart cells and under different activation states is different. So far, a variety of miRNAs that are often enriched in EXOs produced by heart cells have been found, such as miR-320, miR-30a and miR-29b in cardiomyocyte EXOs; miR-451, miR-144, miR-21, miR-146a and so on in cardiac progenitor cells (CPCs) and cardiac globule derived cells (CDCs) EXOs; miR-21 in fibroblast EXOs; and miR-214 and miR-146a in endothelial cell EXOs[1,24,31,32].” Besides, some studies have also reported miRNAs that specifically exist in heart tissue cells, such as miR-208a, miR-1, and miR-195. And we found that, although several studies have provided a general overview of the types and functions of proteins, lipids, and miRNAs commonly found in cardiogenic EXOs, However, no systematic and comprehensive studies about profiles of proteins or miRNAs have been reported by omics deep sequencing. Please see lines 218-241 in section 2 for details.

Point 6: Are there specific miRNA profiles in exosomes isolated from cardiomyocytes, cardiac endothelium etc.? May be start the review with what cell populations are you taking into account when talking about cardiac exosomes otherwise, when reading about MSC-exosomes and adipose tissue-derived exosomes, it is hard to link them to heart exosomes. Also, please, specify - do you review exosomes derived from heart tissue cells or the whole cardiovascular system cells.

Response 6: Some studies have reported that miRNAs exist specifically in cardiac tissue cells, such as miR-1, miR-208a and miR-195. But according to the literature we have reviewed, specific miRNA profiles in exosomes isolated from cardiomyocytes, cardiac endothelium etc. have not be reported. But the relevant research has also made great progress and progress. Please see lines 218-241 in section 2 for more details. In this paper, we review the EXOs derived from heart tissue cells -- including cardiomyocytes, fibroblasts, endothelial cells, cardiac resident stem cells which including cardiac progenitor cells (CPCs) and cardiac globule derived cells (CDCs), cardiac adipose-derived stem cells (ADSCs) and cardiac mesenchymal stem cells (MSCs). Please see lines 144-158 in section 1 for details.

Point 7: What about the other contents of exosomes (proteins and lipids)? What is their profile in cardiac-derived exosomes?

Response 7: There are some common proteins and lipids in exosomes. Similar to the miRNA, according to the literature we have reviewed specific proteins and lipids profiles in cardiac-derived exosomes. Please see lines 193-207 and 232-241 in section 2 for details.

Point 8: If cells can release proteins, why they need to release exosomes (since they also contain proteins)?

Response 8: We think that both direct protein release and exosome release have their own advantages and significance. So they're all life activities that cells need. We have made a supplement to this content in the article. “Therefore, compared with the direct release of cytokines such as proteins, the biggest advantage of EVs for information transfer lies in its strong vesicle membrane, which can protect the contents from the degradation of enzymes in the peripheral circulation or the clearance of the mononuclear phagocytic system[1]. In addition, through EVs system, a variety of information molecules will be transmitted together, so that the transmission efficiency of biological information is greatly increased.” Please see lines 79-85 in section 1 for details.

Point 9: We know that an improvement of angiogenesis is related through secreted VEGF, how cardiac exosomes improves it?

Response 9: Thank you for your important and constructive suggestions. On this topic, we have reviewed the relevant study reports and modified them in the corresponding section of this paper and in Figure 1. Exosomes convey biological information through their molecules, such as miR-126 and HSP20, mediating the phenotypic transformation of target cells, thereby promoting the production and secretion of VEGF, and activating the VEGF receptor 2 (EVGFR2). Ultimately, it promotes angiogenesis. Please see lines 339-350 in section 3.1.3 for details.

Point 10: There were too many problematic parts. If you wish to receive detailed analysis (sentence by sentence), please, send the manuscript with line numbers.

Response 10: Once again, thank you very much for taking time out of your busy schedule to review our article and put forward these very meaningful comments and suggestions. This is very helpful for us to improve our work. As you suggested, we have attached line numbers to this manuscript. We look forward to your review and valuable comments.

Reviewer 2 Report

The manuscript by Haoyang Gao, et al., discusses the pathogenic mechanisms of the role of cardiogenic exosomes that play an important role in regulating the functional state of the heart. The authors dissect mecchanism that almost all the pathological and physiological processes of the heart are inseparable from exosomes in cardio vascular function. Few papers compressively discuss the topics, so this has the potential to add the important to the area.

The manuscript is well-written and the figures are well presented.

Specific comments:

Conclusion: Authors should better highlight the novelty of important findings and, especially, discuss their results in the context of the broad existing literature on the same topic.

English language and style need revision: some grammar mistakes should be amended and several sentences should be rephrased

References: their choice is not always appropriate and should be replaced by new one. Moreover, additional references regarding recent important studies on the epigenetic, immune mechanisms and endocrine part should be included.

Author Response

Response to Reviewer 2 Comments:

Point 1: Conclusion: Authors should better highlight the novelty of important findings and, especially, discuss their results in the context of the broad existing literature on the same topic.

Response 1: Thank you for your critical comments. According to your suggestions, we have made corresponding modifications and supplements in the section ’‘conclusion‘’. To better illustrate the novelty of our findings. Please review the detailed changes in the section conclusion.

Point 2: English language and style need revision: some grammar mistakes should be amended and several sentences should be rephrased.

Response 2: Thank you for your important advice. We have asked native English speakers to revise the language of this article. If you are not satisfied sufficiently, we can revise it again. Please find attached the revised proof of this article.

Point 3: References: their choice is not always appropriate and should be replaced by new one. Moreover, additional references regarding recent important studies on the epigenetic, immune mechanisms and endocrine part should be included.

Response 3: Thank you for your valuable advices. As for references, we added 20 new references, most of which were published in the last three years. We have also added important references you mentioned in these three areas. About epigenetic, we add 2 references and please see lines 119-125 in section 1 for details. About immune mechanisms, we add 2 references and please see lines 125-131 in section 1 for details. About endocrine, we add 1 reference and please see lines 111-114 in section 1 for details.